# FlashSpeech: Efficient Zero-Shot Speech Synthesis

## ABSTRACT

Recent progress in large-scale zero-shot speech synthesis has been significantly advanced by language models and diffusion models. However, the generation process of both methods is slow and computationally intensive. Efficient speech synthesis using a lower computing budget to achieve quality on par with previous work remains a significant challenge. In this paper, we present FlashSpeech, a large-scale zero-shot speech synthesis system with approximately 5% of the inference time compared with previous work. FlashSpeech is built on the latent consistency model and applies a novel adversarial consistency training approach that can train from scratch without the need for a pre-trained diffusion model as the teacher. Furthermore, a new prosody generator module enhances the diversity of prosody, making the rhythm of the speech sound more natural. The generation processes of FlashSpeech can be achieved efficiently with one or two sampling steps while maintaining high audio quality and high similarity to the audio prompt for zero-shot speech generation. Our experimental results demonstrate the superior performance of FlashSpeech. Notably, FlashSpeech can be about 20 times faster than other zero-shot speech synthesis systems while maintaining comparable performance in terms of voice quality and similarity. Furthermore, FlashSpeech demonstrates its versatility by efficiently performing tasks like voice conversion, speech editing, and diverse speech sampling. Audio samples can be found in https://flashspeech.github.io/.

## CCS CONCEPTS

• **Applied computing** → **Sound and music computing**; • **Computing methodologies** → **Natural language generation**.

## KEYWORDS

Zero-Shot Speech Synthesis, Latent Consistency Model, Adversarial Training

## 1 INTRODUCTION

In recent years, the landscape of speech synthesis has been transformed by the advent of large-scale generative models. Consequently, the latest research efforts have achieved notable advancements in zero-shot speech synthesis systems by significantly increasing the size of both datasets and models. Zero-shot speech synthesis, such as text-to-speech (TTS), voice conversion (VC) and Editing, aims to generate speech that incorporates unseen

**Unpublished working draft. Not for distribution.**

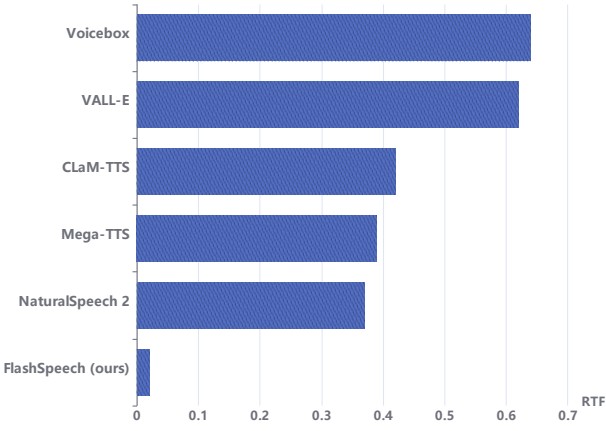

**Figure 1: The inference time comparisons of different zero-shot speech synthesis systems using the real-time factor (RTF).**

speaker characteristics from a reference audio segment during inference, without the need for additional training. Current advanced zero-shot speech synthesis systems typically leverage language models (LMs) [20, 22, 39, 59, 61, 63, 66] and diffusion-style models [18, 23, 26, 52] for in-context speech generation on the large-scale dataset. However, the generation process of these methods needs a long-time iteration. For example, VALL-E [59] builds on the language model to predict 75 audio token sequences for a 1-second speech, in its first-stage autoregressive (AR) token sequence generation. When using a non-autoregressive (NAR) latent diffusion model [47] based framework, NaturalSpeech 2 [52] still requires 150 sampling steps. As a result, although these methods can produce human-like speech, they require significant computational time and cost. Some efforts have been made to accelerate the generation process. Voicebox [26] adopts flow-matching [29] so that fewer sampling steps (NFE[1]: 64) can be achieved because of the optimal transport path. ClaM-TTS [22] proposes a mel-codec with a superior compression rate and a latent language model that generates a stack of tokens at once. Although the slow generation speed issue has been somewhat alleviated, the inference speed is still far from satisfactory for practical applications. Moreover, the substantial computational time of these approaches leads to significant computational cost overheads, presenting another challenge.

The fundamental limitation of speech generation stems from the intrinsic mechanisms of language models and diffusion models, which require considerable time either auto-regressively or through a large number of denoising steps. Hence, the primary objective of this work is to accelerate inference speed and reduce computational costs while preserving generation quality at levels comparable to

---

[1]NFE: number of function evaluations.

the prior research. In this paper, we propose FlashSpeech as the next step towards efficient zero-shot speech synthesis. To address the challenge of slow generation speed, we leverage the latent consistency model (LCM) [32], a recent advancement in generative models. Building upon the previous non-autoregressive TTS system [52], we adopt the encoder of a neural audio codec to convert speech waveforms into latent vectors as the training target for our LCM. To train this model, we propose a novel technique called adversarial consistency training, which utilizes the capabilities of pre-trained speech language models [1, 6, 11] as discriminators. This facilitates the transfer of knowledge from large pre-trained speech language models to speech generation tasks, efficiently integrating adversarial and consistency training to improve performance. The LCM is conditioned on prior vectors obtained from a phoneme encoder, a prompt encoder, and a prosody generator. Furthermore, we demonstrate that our proposed prosody generator leads to more diverse expressions and prosody while preserving stability.

Our contributions can be summarized as follows:

- We propose FlashSpeech, an efficient zero-shot speech synthesis system that generates voice with high audio quality and speaker similarity in zero-shot scenarios.
- We introduce adversarial consistency training, a novel combination of consistency and adversarial training leveraging pre-trained speech language models, for training the latent consistency model from scratch, achieving speech generation in one or two steps.
- We propose a prosody generator module that enhances the diversity of prosody while maintaining stability.
- FlashSpeech significantly outperforms strong baselines in audio quality and matches them in speaker similarity. Remarkably, it achieves this at a speed approximately 20 times faster than comparable systems, demonstrating unprecedented efficiency.

## 2 RELATED WORK

### 2.1 Large Scale Speech Synthesis

Motivated by the success of the large language model, the speech research community has recently shown increasing interest in scaling the sizes of model and training data to bolster generalization capabilities, producing natural speech with diverse speaker identities and prosody under zero-shot settings. The pioneering work is VALL-E [59], which adopts the Encodec [8] to discretize the audio waveform into tokens. Therefore, a language model can be trained via in-context learning that can generate the target utterance where the style is consistent with prompt utterance. However, generating audio in such an autoregressive manner [39, 61]can lead to unstable prosody, word skipping, and repeating issues [44, 52, 57]. To ensure the robustness of the system, non-autoregressive methods such as NaturalSpeech2 [52] and Voicebox [26] utilize diffusion-style model (VP-diffusion [55] or flow-matching [29]) to learn the distribution of a continuous intermediate vector such as mel-spectrogram or latent vector of codec. Both LM-based methods [67] and diffusion-based methods show superior performance in speech generation tasks. However, their generation is slow due to the iterative computation. Considering that many speech generation scenarios require real-time inference and low computational costs, we employ the latent

consistency model for large-scale speech generation that inference with one or two steps while maintaining high audio quality.

### 2.2 Acceleration of Speech Synthesis

Since early neural speech generation models [57] use autoregressive models such as Tacotron [60] and TransformerTTS [27], causing slow inference speed, with $O(N)$ computation, where $N$ is the sequence length. To address the slow inference speed, FastSpeech [44, 45] proposes to generate a mel-spectrogram in a non-autoregressive manner. However, these models [46] result in blurred and over-smoothed mel-spectrograms due to the regression loss they used and the capability of modeling methods. To further enhance the speech quality, diffusion models are utilized [13, 40, 41] which increase the computation to $O(T)$, where T is the diffusion steps. Therefore, distillation techniques [33] for diffusion-based methods such as CoMoSpeech [64], CoMoSVC [31] and Reflow-TTS [10] emerge to reduce the sampling steps back to $O(1)$, but require additional pre-trained diffusion as the teacher model. Unlike previous distillation techniques, which require extra training for the diffusion model as a teacher and are limited by its performance, our proposed adversarial consistency training technique can directly train from scratch, significantly reducing training costs. In addition, previous acceleration methods only validate speaker-limited recording-studio datasets with limited data diversity. To the best of our knowledge, FlashSpeech is the first work that reduces the computation of a large-scale speech generation system back to $O(1)$.

### 2.3 Consistency Model

The consistency model is proposed in [53, 54] to generate high-quality samples by directly mapping noise to data. Furthermore, many variants [21, 25, 30, 51] are proposed to further increase the generation quality of images. The latent consistency model is proposed by [32] which can directly predict the solution of PF-ODE in latent space. However, the original LCM employs consistency distillation on the pre-trained latent diffusion model (LDM) which leverages large-scale off-the-shelf image diffusion models [47]. Since there are no pre-trained large-scale TTS models in the speech community, and inspired by the techniques [21, 25, 30, 51, 53], we propose the novel adversarial consistency training method which can directly train the large-scale latent consistency model from scratch utilizing the large pre-trained speech language model [1, 6, 11] such as WavLM for speech generation.

## 3 FLASHSPEECH

### 3.1 Overview

Our work is dedicated to advancing the speech synthesis efficiency, achieving $O(1)$ computation cost while maintaining comparable performance to prior studies that require $O(T)$ or $O(N)$ computations. The framework of the proposed method, FlashSpeech, is illustrated in Fig. 2. FlashSpeech integrates a neural codec, an encoder for phonemes and prompts, a prosody generator, and an LCM, which are utilized during both the training and inference stages. Exclusively during training, a conditional discriminator is employed. FlashSpeech adopts the in-context learning paradigm [59], initially segmenting the latent vector $\mathbf{z}$, extracted from the

Figure 2: Overall architecture of FlashSpeech. Our FlashSpeech consists of a codec encoder/decoder and a latent consistency model conditioned on feature from a phoneme and $z_{prompt}$ encoder and a prosody generator. A discriminator is used during training.

codec, into $\mathbf{z}_{target}$ and $\mathbf{z}_{prompt}$. Subsequently, the phoneme and $\mathbf{z}_{prompt}$ are processed through the encoder to produce the hidden feature. A prosody generator then predicts pitch and duration based on the hidden feature. The pitch and duration embeddings are combined with the hidden feature and inputted into the LCM as the conditional feature. The LCM model is trained from scratch using adversarial consistency training. After training, FlashSpeech can achieve efficient generation within one or two sampling steps.

## 3.2 Latent Consistency Model

The consistency model [54] is a new family of generative models that enables one-step or few-step generation. Let us denote the data distribution by $p_{\text{data}}(\mathbf{x})$. The core idea of the consistency model is to learn the function that maps any points on a trajectory of the PF-ODE to that trajectory's origin, which can be formulated as:

$$f(\mathbf{x}_\sigma, \sigma) = \mathbf{x}_{\sigma_{min}} \qquad (1)$$

where $f(\cdot, \cdot)$ is the consistency function and $\mathbf{x}_\sigma$ represents the data $\mathbf{x}$ perturbed by adding zero-mean Gaussian noise with standard deviation $\sigma$. $\sigma_{min}$ is a fixed small positive number. Then $\mathbf{x}_{\sigma_{min}}$ can then be viewed as an approximate sample from the data distribution $p_{\text{data}}(\mathbf{x})$. To satisfy property in equation (1), following [54], we parameterize the consistency model as

$$f_\theta(\mathbf{x}_\sigma, \sigma) = c_{\text{skip}}(\sigma)\mathbf{x} + c_{\text{out}}(\sigma)F_\theta(\mathbf{x}_\sigma, \sigma) \qquad (2)$$

where $f_\theta$ is to estimate consistency function $f$ by learning from data, $F_\theta$ is a deep neural network with parameter $\theta$, $c_{\text{skip}}(\sigma)$ and $c_{\text{out}}(\sigma)$ are are differentiable functions with $c_{\text{skip}}(\sigma_{min}) = 1$ and $c_{\text{out}}(\sigma_{min}) = 0$ to ensure boundary condition. A valid consistency model should satisfy the self-consistency property [54]

$$f_\theta(\mathbf{x}_\sigma, \sigma) = f_\theta(\mathbf{x}_{\sigma'}, \sigma'), \quad \forall \sigma, \sigma' \in [\sigma_{min}, \sigma_{max}]. \qquad (3)$$

where $\sigma_{max} = 80$ and $\sigma_{min} = 0.002$ following [19, 53, 54]. Then the model can generate samples in one step by evaluating

$$\mathbf{x}_{\sigma_{min}} = f_\theta(\mathbf{x}_{\sigma_{max}}, \sigma_{max}) \qquad (4)$$

from distribution $\mathbf{x}_{\sigma_{max}} \sim \mathcal{N}(0, \sigma_{max}^2 \mathbf{I})$.

As we apply a consistency model on the latent space of audio, we use the latent features $z$ which are extracted prior to the residual quantization layer of the codec,

$$\mathbf{z} = CodecEncoder(\mathbf{y}) \qquad (5)$$

where $\mathbf{y}$ is the speech waveform. Furthermore, we add the feature from the prosody generator and encoder as the conditional feature $c$, our objective has changed to achieve

$$f_\theta(\mathbf{z}_\sigma, \sigma, c) = f_\theta(\mathbf{z}_{\sigma'}, \sigma', c) \quad \forall \sigma, \sigma' \in [\sigma_{min}, \sigma_{max}]. \qquad (6)$$

During inference, the synthesized waveform $\hat{y}$ is transformed from $\hat{z}$ via the codec decoder. The predicted $\hat{z}$ is obtained by one sampling step

$$\hat{\mathbf{z}} = f_\theta(\epsilon * \sigma_{max}, \sigma_{max}) \qquad (7)$$

or two sampling steps

$$\hat{\mathbf{z}}_{\text{inter}} = f_\theta(\epsilon * \sigma_{max}, \sigma_{max}) \qquad (8)$$
$$\hat{\mathbf{z}} = f_\theta(\hat{\mathbf{z}}_{\text{inter}} + \epsilon * \sigma_{\text{inter}}, \sigma_{\text{inter}}) \qquad (9)$$

where $\hat{\mathbf{z}}_{\text{inter}}$ means the intermediate step, $\sigma_{\text{inter}}$ is set to 2 empirically. $\epsilon$ is sampled from a standard Gaussian distribution.

## 3.3 Adversarial Consistency Training

A major drawback of the LCM[32] is that it needs to pre-train a diffusion-based teacher model in the first stage, and then perform distillation to produce the final model. This would make the training process complicated, and the performance would be limited as a result of the distillation. To eliminate the reliance on the teacher

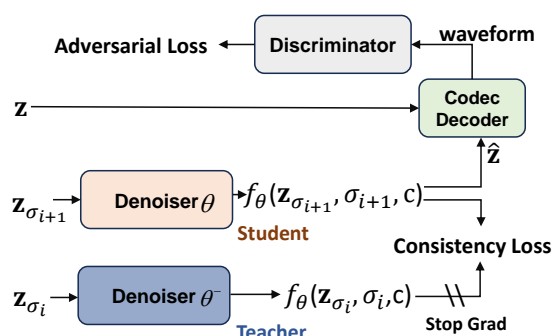

Figure 3: An illustration of adversarial consistency training.

model training, in this paper, we propose a novel adversarial consistency training method to train LCM from scratch. Our training procedure is outlined in Fig. 3, which has three parts:

*3.3.1 Consistency Training.* To achieve the property in equation (3), we adopt following consistency loss

$$\mathcal{L}_{ct}^N(\theta, \theta^-) = \mathbb{E}[\lambda(\sigma_i)d(f_\theta(z_{i+1}, \sigma_{i+1}, c), f_{\theta^-}(z_i, \sigma_i, c))]. \quad (10)$$

where $\sigma_i$ represents the noise level at discrete time step $i$, $d(\cdot, \cdot)$ is the distance function, $f_\theta(z_{i+1}, \sigma_{i+1}, c)$ and $f_{\theta^-}(z_i, \sigma_i, c)$ are the student with the higher noise level and the teacher with the lower noise level, respectively. The discrete time steps denoted as $\sigma_{min} = \sigma_0 < \sigma_1 < \cdots < \sigma_N = \sigma_{max}$ are divided from the time interval $[\sigma_{min}, \sigma_{max}]$, where the discretization curriculum $N$ increases correspondingly as the number of training steps grows

$$N(k) = \min(s_0 2^{\lfloor \frac{k}{K'} \rfloor}, s_1) + 1 \quad (11)$$

where $K' = \left\lfloor \frac{K}{\log_2 \lfloor s_1/s_0 \rfloor + 1} \right\rfloor$, $k$ is the current training step and $K$ is the total training steps. $s_1$ and $s_0$ are hyperparameters to control the size of $N(k)$. The distance function $d(\cdot, \cdot)$ uses the Pseudo-Huber metric [3]

$$d(x, y) = \sqrt{\|x - y\|^2 + a^2} - a, \quad (12)$$

where $a$ is an adjustable constant, making the training more robust to outliers as it imposes a smaller penalty for large errors than $\ell_2$ loss. The parameters $\theta^-$ of teacher model are

$$\theta^- \longleftarrow stopgrad(\theta), \quad (13)$$

which are identical to the student parameters $\theta$. This approach [53] has been demonstrated to improve sample quality of previous strategies that employ varying decay rates [54]. The weighting function refers to

$$\lambda(\sigma_i) = \frac{1}{\sigma_{i+1} - \sigma_i} \quad (14)$$

which emphasizes the loss of smaller noise levels. LCM through consistency training can generate speech with acceptable quality in a few steps, but it still falls short of previous methods. Therefore, to further enhance the quality of the generated samples, we integrate adversarial training.

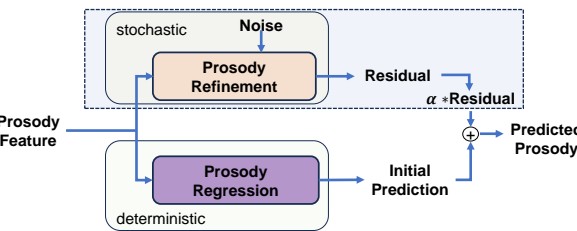

Figure 4: An illustration of prosody generator.

*3.3.2 Adversarial Training.* For the adversarial objective, the generated samples $\hat{z} \leftarrow f_\theta(z_\sigma, \sigma, c)$ and real samples $z$ are passed to the discriminator $D_\eta$ which aims to distinguish between them, where $\eta$ refers to the trainable parameters. Thus, we employ adversarial training loss

$$\mathcal{L}_{adv}(\theta, \eta) = \mathbb{E}_z[\log D_\eta(z)] + \mathbb{E}_\sigma \mathbb{E}_{z_\sigma}[\log(1 - D_\eta(f_\theta(z_\sigma, \sigma, c)))]. \quad (15)$$

In this way, the error signal from the discriminator guides $f_\theta$ to produce more realistic outputs. For details, we use a frozen pretrained speech language model $SLM$ and a trainable lightweight discriminator head $D_{head}$ to build the discriminator. Since the current $SLM$ is trained on the speech waveform, we covert both $z$ and $\hat{z}$ to ground truth waveform and predicted waveform using the codec decoder. To further increase the similarity between prompt audio and generated audio, our discriminator is conditioned on the prompt audio feature. This prompt feature $F_{prompt}$ is extracted using $SLM$ on prompt audio and applies average pooling on the time axis. Therefore,

$$D_\eta = D_{head}(F_{prompt} \odot F_{gt}, F_{prompt} \odot F_{pred}) \quad (16)$$

where $F_{gt}$ and $F_{pred}$ refer to feature extracted through $SLM$ for ground truth waveform and predicted waveform. The discriminator head consists of several 1D convolution layers. The input feature of the discriminator is conditioned on $F_{prompt}$ via projection [36].

*3.3.3 Combined Together.* Since there is a large gap on the loss scale between consistency loss and adversarial loss, it can lead to instability and failure in training. Therefore, we follow [9] to compute the adaptive weight with

$$\lambda_{adv} = \frac{\|\nabla_{\theta_L} \mathcal{L}_{ct}^N(\theta, \theta^-)\|}{\|\nabla_{\theta_L} \mathcal{L}_{adv}(\theta, \eta)\|} \quad (17)$$

where $\theta_L$ is the last layer of the neural network in LCM. The final loss of training LCM is defined as $\mathcal{L}_{ct}^N(\theta, \theta^-) + \lambda_{adv}\mathcal{L}_{adv}(\theta, \eta)$. This adaptive weighting significantly stabilizes the training by balancing the gradient scale of each term.

## 3.4 Prosody Generator

*3.4.1 Analysis of the prediction of prosody.* Previous regression methods for prosody prediction [44, 52], due to their deterministic mappings and assumptions of unimodal distribution, often fail to capture the inherent diversity and expressiveness of human speech prosody. This leads to predictions that lack variation and can appear over-smoothed. On the other hand, diffusion methods [26, 28] for prosody prediction offer a promising alternative by providing greater prosody diversity. However, they come with challenges

regarding stability, and the potential for unnatural prosody. Additionally, the iterative inference process in DMs requires a significant number of sampling steps that may also hinder real-time application. Meanwhile, LM-based methods [15, 59] also need a long time for inference. To alleviate these issues, our prosody generator consists of a prosody regression module and a prosody refinement module to enhance the diversity of prosody regression results with efficient one-step consistency model sampling.

*3.4.2 Prosody refinement via consistency model.* As shown in 4, our prosody generator consists of two parts which are prosody regression and prosody refinement. We first train the prosody regression module to get a deterministic output. Next, we freeze the parameters of the prosody regression module and use the residual of ground truth prosody and deterministic predicted prosody as the training target for prosody refinement. We adopt a consistency model as a prosody refinement module. The conditional feature of the consistency model is the feature from prosody regression before the final projection layer. Thus, the residual from a stochastic sampler refines the output of a deterministic prosody regression and produces a diverse set of plausible prosody under the same transcription and audio prompt. One option for the final prosody output $p_{\text{final}}$ can be represented as:

$$p_{\text{final}} = p_{\text{res}} + p_{\text{init}}, \tag{18}$$

where $p_{\text{final}}$ denotes the final prosody output, $p_{\text{res}}$ represents the residual output from the prosody refinement module, capturing the variations between the ground truth prosody and the deterministic prediction, $p_{\text{init}}$ is the initial deterministic prosody prediction from the prosody regression module. However, this formulation may negatively affect prosody stability, a similar observation is found in [26, 58]. More specifically, higher diversity may cause less stability and sometimes produce unnatural prosody. To address this, we introduce a control factor $\alpha$ that finely tunes the balance between stability and diversity in the prosodic output:

$$p_{\text{final}} = \alpha p_{\text{res}} + p_{\text{init}} \tag{19}$$

where $\alpha$ is a scalar value ranging between 0 and 1. This adjustment allows for controlled incorporation of variability into the prosody, mitigating issues related to stability while still benefiting from the diversity offered by the prosody refinement module.

## 3.5 Applications

This section elaborates on the practical applications of FlashSpeech. We delve into its deployment across various tasks such as zero-shot TTS, speech editing, voice conversion, and diverse speech sampling. All the sample audios of applications are available on the demo page.

*3.5.1 Zero-Shot TTS.* Given a target text and reference audio, we first convert the text to phoneme using g2p (grapheme-to-phoneme conversion). Then we use the codec encoder to convert the reference audio into $\mathbf{z}_{prompt}$. Speech can be synthesized efficiently through FlashSpeech with the phoneme input and $\mathbf{z}_{prompt}$, achieving high-quality text-to-speech results without requiring pre-training on the specific voice.

*3.5.2 Voice Conversion.* Voice conversion aims to convert the source audio into the target audio using the speaker's voice of the reference audio. Following [43, 52], we first apply the reverse of ODE to diffuse the source audio into a starting point that still maintains some information in the source audio. After that, we run the sampling process from this starting point with the reference audio as $\mathbf{z}_{prompt}$ and condition $c$. The condition $c$ uses the phoneme and duration from the source audio and the pitch is predicted by the prosody generator. This method allows for zero-shot voice conversion while preserving the linguistic content of the source audio, and achieving the same timbre as the reference audio.

*3.5.3 Speech Editing.* Given the speech, the original transcription, and the new transcription, we first use MFA (Montreal Forced Aligner) to align the speech and the original transcription to get the duration of each word. Then we remove the part that needs to be edited to construct the reference audio. Next, we use the new transcription and reference to synthesize new speech. Since this task is consistent with the in-context learning, we can concatenate the remaining part of the raw speech and the synthesized part as the final speech, thus enabling precise and seamless speech editing.

*3.5.4 Diverse Speech Sampling.* FlashSpeech leverages its inherent stochasticity to generate a variety of speech outputs under the same conditions. By employing stochastic sampling in its prosody generation and LCM, FlashSpeech can produce diverse variations in pitch, duration, and overall audio characteristics from the same phoneme input and audio prompt. This feature is particularly useful for generating a wide range of speech expressions and styles from a single input, enhancing applications like voice acting, synthetic voice variation for virtual assistants, and more personalized speech synthesis. In addition, the synthetic data via speech sampling can also benefit other tasks such as ASR [48].

## 4 EXPERIMENT

In the experimental section, we begin by introducing the datasets and the configurations for training in our experiments. Following this, we show the evaluation metrics and demonstrate the comparative results against various zero-shot TTS models. Subsequently, ablation studies are conducted to test the effectiveness of several design choices. Finally, we also validate the effectiveness of other tasks such as voice conversion. We show our speech editing and diverse speech sampling results on our demo page.

## 4.1 Experimental Settings

*4.1.1 Data and Preprocessing.* We use the English subset of Multilingual LibriSpeech (MLS) [42], including 44.5k hours of transcribed audiobook data and it contains 5490 distinct speakers. The audio data is resampled at a frequency of 16kHz. The input text is transformed into a sequence of phonemes through grapheme-to-phoneme conversion [56] and then we use our internal alignment tool aligned with speech to obtain the phoneme-level duration. We adopt a hop size of 200 for all frame-level features. The pitch sequence is extracted using PyWorld[2]. we adopt Encodec [8] as our audio codec. We use a modified version [3] and train it on MLS. We

---

[2]https://github.com/JeremyCCHsu/Python-Wrapper-for-World-Vocoder
[3]https://github.com/yangdongchao/UniAudio/tree/main/codec

**Table 1: The evaluation results for FlashSpeech and the baseline methods on LibriSpeech testclean. ★ means the evaluation is conducted with 1 NVIDIA V100 GPU. ◇ means the device is not available. Abbreviation: MLS (Multilingual LibriSpeech [42]),G (GigaSpeech [4]), L(LibriTTS-R [24]), V(VCTK [62]), LJ(LJSpeech [12]), W(WenetSpeech [65]).**

| Model | Training Data | RTF ↓ | Sim-O ↑ | Sim-R ↑ | WER ↓ | CMOS ↑ | SMOS (↑) |
|---|---|---|---|---|---|---|---|
| GroundTruth | - | - | 0.68 | - | 1.9 | 0.11 | 4.39 |
| VALL-E reproduce [59] | Librilight | 0.62 ◇ | 0.47 | 0.51 | 6.1 | -0.48 | 4.11 |
| NaturalSpeech 2 [52] | MLS | 0.37 (NFE:150) ★ | **0.53** | **0.60** | **1.9** | -0.31 | 4.20 |
| Voicebox reproduce [26] | Librilight | 0.66 (NFE:64) ◇ | 0.48 | 0.50 | 2.1 | -0.58 | 3.95 |
| Mega-TTS [17] | G+W | 0.39 ◇ | - | - | 3.0 | - | - |
| CLaM-TTS [22] | MLS+G+L+V+LJ | 0.42 ◇ | 0.50 | 0.54 | 5.1 | - | - |
| FlashSpeech (ours) | MLS | **0.02** (NFE: 2) ★ | 0.52 | 0.57 | 2.7 | **0.00** | **4.29** |

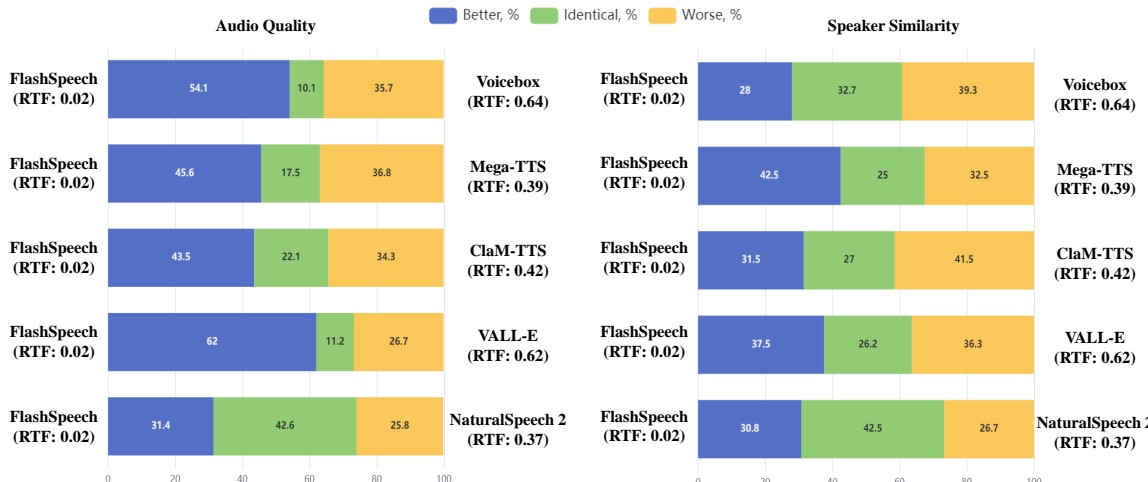

**Figure 5: User preference study. We compare the audio quality and speaker similarity of FlashSpeech against baselines with their official demo.**

use the dense features extracted before the residual quantization layer as our latent vector $z$.

*4.1.2 Training Details .* Our training consists of two stages, in the first stage we train LCM and the prosody regression part. We use 8 H800 80GB GPUs with a batch size of 20k frames of latent vectors per GPU for 650k steps. We use the AdamW optimizer with a learning rate of 3e-4, warm up the learning rate for the first 30k updates and then linear decay it. We deactivate adversarial training with $\lambda_{adv} = 0$ before 600K training iterations. For hyper-parameters, we set $a$ in Equation (12) to 0.03. In equation (10), $\sigma_i = \left(\sigma_{\min}^{1/\rho} + \frac{i-1}{N(k)-1}\left(\sigma_{\max}^{1/\rho} - \sigma_{\min}^{1/\rho}\right)\right)^{\rho}$, where $i \in [1, N(k)]$, $\rho = 7$, $\sigma_{\min} = 0.002$, $\sigma_{\max} = 80$. For N(k) in Equation (11), we set $s_0 = 10, s_1 = 1280, K = 600k$. After 600k steps, we activate adversarial loss, and N(k) can be considered as fixed to 1280. We crop the waveform length fed into the discriminator into minimum waveform length in a minibatch. In addition, the weight of the feature extractor WavLM and the codec decoder are frozen.

In the second stage, we train 150k steps for the prosody refinement module with consistency training in Equation (10). Different from the above setting, we empirically set $s_1 = 160, K = 150k$.

During training, only the weight of the prosody refinement part is updated.

*4.1.3 Model Details .* The model structures of the prompt encoder and phoneme encoder are follow[52]. The neural function part in LCM is almost the same as the [52]. We rescale the sinusoidal position embedding in the neural function part by a factor of 1000. As for the prosody generator, we adopt 30 non-casual wavenet [37] layers for the neural function part in the prosody refinement module and the same configurations for prosody regression parts in [52]. And we set $\alpha = 0.2$ for the prosody refinement module empirically. For the discriminator's head, we stack 5 convolutional layers with weight normalization [50] for binary classification.

## 4.2 Evaluation Metrics

We use both objective and subjective evaluation metrics, including

- **RTF**: Real-time-factor (RTF) measures the time taken for the system to generate one second of speech. This metric is crucial for evaluating the efficiency of our system, particularly for applications requiring real-time processing. We measure

the time of our system end-to-end on an NVIDIA V100 GPU following [52].

- **Sim-O and Sim-R**: These metrics assess the speaker similarity. Sim-R measures the objective similarity between the synthesized speech and the reconstruction reference speech through the audio codec, using features embedding extracted from the pre-trained speaker verification model [22, 59][4]. Sim-O is calculated with the original reference speech. Higher scores in Sim-O and Sim-R indicate a higher speaker similarity.
- **WER (Word Error Rate)**: To evaluate the accuracy and clarity of synthesized speech from the TTS system, we employ the Automatic Speech Recognition (ASR) model [59] [5] to transcribe generated audio. The discrepancies between these transcriptions and original texts are quantified using the Word Error Rate (WER), a crucial metric indicating intelligibility and robustness.
- **CMOS, SMOS, UTMOS**: we rank the comparative mean option score (CMOS) and similarity mean option score (SMOS) using mturk. The prompt for CMOS refers to 'Please focus on the audio quality and naturalness and ignore other factors.'. The prompt for SMOS refers to 'Please focus on the similarity of the speaker to the reference, and ignore the differences of content, grammar or audio quality.' Each audio has been listened to by at least 10 listeners. UTMOS [49] is a Speech MOS predictor[6] to measure the naturalness of speech. We use it in ablation studies which reduced the cost for evaluation.
- **Prosody JS Divergence**: To evaluate the diversity and accuracy of the prosody prediction in our TTS system, we include the Prosody JS Divergence metric. This metric employs the Jensen-Shannon (JS) divergence [35] to quantify the divergence between the predicted and ground truth prosody feature distributions. Prosody features, including pitch, and duration, are quantized and their distributions in both synthesized and natural speech are compared. Lower JS divergence values indicate closer similarity between the predicted prosody features and those of the ground truth, suggesting a higher diversity of the synthesized speech.

## 4.3 Experimental Results on Zero-shot TTS

Following [59], We employ LibriSpeech [38] test-clean for zero-shot TTS evaluation. We adopt the cross-sentence setting in [59] that we randomly select 3-second clips as prompts from the same speaker's speech. The results are summarized in table 1 and figure 5.

### 4.3.1 Evaluation Baselines.

- **VALL-E [59]**: VALL-E predicts codec tokens using both AR and NAR models. RTF[7] is obtained from [22, 26]. We use our reproduced results for MOS, Sim, and WER. Additionally, we do a preference test with their official demo.

- **Voicebox [26]**: Voicebox uses flow-matching to predict maksed mel-spectrogram. RTF is from the original paper. We use our reproduced results for MOS, Sim, and WER. We also implement a preference test with their official demo.
- **NaturalSpeech2 [52]**: NaturalSpeech2 uses a latent diffusion model to predict latent features of codec. The RTF is from the original paper. the Sim, WER and samples for MOS are obtained through communication with the authors. We also do a preference test with their official demo.
- **Mega-TTS [17][8]**: Mega-TTS uses both language model and GAN to predict mel-spectrogram. We obtain RTF from mobilespeech [14] and WER from the original paper. We do a preference test with their official demo.
- **ClaM-TTS [22]**: ClaM-TTS uses the AR model to predict mel codec tokens. We obtain the objective evaluation results from the original paper and do a preference test with their official demo.

### 4.3.2 Generation Quality.
FlashSpeech stands out significantly in terms of speaker quality, surpassing other baselines in both CMOS and audio quality preference tests. Notably, our method closely approaches ground truth recordings, underscoring its effectiveness. These results affirm the superior quality of FlashSpeech in speech synthesis. our method.

### 4.3.3 Generation Similarity.
Our evaluation of speaker similarity utilizes Sim, SMOS, and speaker similarity preference tests, where our methods achieve 1st, 2nd, and 3rd place rankings, respectively. These findings validate our methods' ability to achieve comparable speaker similarity to other methods. Despite our training data (MLS) containing approximately 5k speakers, fewer than most other methods (e.g., Librilight with about 7k speakers or self-collected data), we believe that increasing the number of speakers in our methods can further enhance speaker similarity.

### 4.3.4 Robustness.
Our methods achieve a WER of 2.7, placing them in the first echelon. This is due to the non-autoregressive nature of our methods, which ensures robustness.

### 4.3.5 Generation Speed.
FlashSpeech achieves a remarkable approximately 20x faster inference speed compared to previous work. Considering its excellent audio quality, robustness, and comparable speaker similarity, our method stands out as an efficient and effective solution in the field of large-scale speech synthesis.

## 4.4 Ablation Studies

### 4.4.1 Ablation studies of LCM.
We explored the impact of different pre-trained models in adversarial training on UTMOS and Sim-O. As shown in the table 2, the baseline, which employs consistency training alone, achieved a UTMOS of 3.62 and a Sim-O of 0.45. Incorporating adversarial training using wav2vec2-large[9], hubert-large[10], and wavlm-large[11] as discriminators significantly improved both UTMOS and Sim-O scores. Notably, the application of adversarial training with Wavlm-large achieved the highest scores

---

[4]https://github.com/microsoft/UniSpeech/tree/main/downstreams/speaker_verification
[5]https://huggingface.co/facebook/hubert-large-ls960-ft
[6]https://github.com/tarepan/SpeechMOS
[7]In ClaM-TTS and Voicebox, they report the inference time for generating 10 seconds of speech. Therefore, we divide by 10 to obtain the time for generating 1 second of speech (RTF).

[8]Since we do not find any audio samples for Mega-TTS2 [16] under the 3-second cross-sentence setting, we are not able to compare with them.
[9]https://huggingface.co/facebook/wav2vec2-large
[10]https://huggingface.co/facebook/hubert-large-ll60k
[11]https://huggingface.co/microsoft/wavlm-large

**Table 2: The ablation study of discriminator design.**

| Method | UTMOS ↑ | Sim-O ↑ |
|---|---|---|
| Consistency training baseline | 3.62 | 0.45 |
| + Adversarial training (Wav2Vec2-large) | 3.92 | 0.50 |
| + Adversarial training (Hubert-large) | 3.83 | 0.47 |
| + Adversarial training (Wavlm-large) | **4.00** | **0.52** |
|   - prompt projection | 3.97 | 0.51 |

**Table 3: The ablation study of sampling steps for LCM**

| NFE | UTMOS ↑ | Sim-O ↑ |
|---|---|---|
| 1 | 3.99 | 0.51 |
| 2 | **4.00** | **0.52** |
| 4 | 3.91 | 0.51 |

(UTMOS: 4.00, Sim-O: 0.52), underscoring the efficacy of this pre-trained model in enhancing the quality and speaker similarity of synthesized speech. Additionally, without using the audio prompt's feature as a condition the discriminator shows a slight decrease in performance (UTMOS: 3.97, Sim-O: 0.51), highlighting the importance of conditional features in guiding the adversarial training process.

As shown in table 3, the effect of sampling steps (NFE) on UTMOS and Sim-O revealed that increasing NFE from 1 to 2 marginally improves UTMOS (3.99 to 4.00) and Sim-O (0.51 to 0.52). However, further increasing to 4 sampling steps slightly reduced UTMOS to 3.91 due to the accumulation of score estimation errors [5, 34]. Therefore, we use 2 steps as the default setting for LCM.

*4.4.2 Ablation studies of Prosody Generator.* In this part, we investigated the effects of a control factor, denoted as $\alpha$, on the prosodic features of pitch and duration in speech synthesis, by setting another influencing factor to zero. Our study specifically conducted an ablation analysis to assess how $\alpha$ influences these features, emphasizing its critical role in balancing stability and diversity within our framework's prosodic outputs.

Table 4 elucidates the effects of varying $\alpha$ on the pitch component. With $\alpha$ set to 0, indicating no inclusion of the residual output from prosody refinement, we observed a Pitch JSD of 0.072 and a WER of 2.8. A slight modification to $\alpha = 0.2$ resulted in a reduced Pitch JSD of 0.067, maintaining the same WER. Notably, setting $\alpha$ to 1, fully incorporating the prosody refinement's residual output, further decreased the Pitch JSD to 0.063, albeit at the cost of increased WER to 3.7, suggesting a trade-off between prosody diversity and speech intelligibility.

Similar trends in table 5 are observed in the duration component analysis. With $\alpha = 0$, the Duration JSD was 0.0175 with a WER of 2.8. Adjusting $\alpha$ to 0.2 slightly improved the Duration JSD to 0.0168, without affecting WER. However, fully embracing the refinement module's output by setting $\alpha = 1$ yielded the most significant improvement in Duration JSD to 0.0153, which, similar to pitch analysis, came with an increased WER of 3.9. The results underline the delicate balance required in tuning $\alpha$ to optimize between

**Table 4: The ablation study of control factor for pitch**

| $\alpha$ | Pitch JSD ↓ | WER↓ |
|---|---|---|
| 0 | 0.072 | 2.8 |
| 0.2 | 0.067 | 2.8 |
| 1 | 0.063 | 3.7 |

**Table 5: The ablation study of control factor for duration**

| $\alpha$ | Duration JSD ↓ | WER ↓ |
|---|---|---|
| 0 | 0.0175 | 2.8 |
| 0.2 | 0.0168 | 2.8 |
| 1 | 0.0153 | 3.9 |

**Table 6: Voice Conversion**

| Method | CMOS ↑ | SMOS ↑ | Sim-O ↑ |
|---|---|---|---|
| YourTTS [2] | -0.16 | 3.26 | 0.23 |
| DDDM-VC [7] | -0.28 | 3.43 | 0.28 |
| Ours | **0.00** | **3.50** | **0.35** |

diversity and stability of prosody without compromising speech intelligibility.

## 4.5 Evaluation Results for Voice Conversion

In this section, we present the evaluation results of our voice conversion system, FlashSpeech, in comparison with state-of-the-art methods, including YourTTS [12] [2] and DDDM-VC [13] [7]. We conduct the experiments with their official checkpoints in our internal test set.

Our system outperforms both YourTTS and DDDM-VC in terms of CMOS, SMOS and Sim-O, demonstrating its capability to produce converted voices with high quality and similarity to the target speaker. These results confirm the effectiveness of our FlashSpeech approach in voice conversion tasks.

## 4.6 Conclusions and Future Work

In this paper, we presented FlashSpeech, a novel speech generation system that significantly reduces computational costs while maintaining high-quality speech output. Utilizing a novel adversarial consistency training method and an LCM, FlashSpeech outperforms existing zero-shot TTS systems in efficiency, achieving speeds about 20 times faster without compromising on voice quality, similarity, and robustness. In the future, we aim to further refine the model to improve the inference speed and reduce computational demands. In addition, we will expand the data scale and enhance the system's ability to convey a broader range of emotions and more nuanced prosody. For future applications, FlashSpeech can be integrated for real-time interactions in applications such as virtual assistants and educational tools.

---

[12]https://github.com/coqui-ai/TTS
[13]https://github.com/hayeong0/DDDM-VC

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
