# OpenReview forum: "FlashSpeech: Efficient Zero-Shot Speech Synthesis"
_acmmm.org/ACMMM/2024/Conference — MM2024 Poster_

### Official Review · Reviewer_ekAV · 2024-05-05

**Rating:** 3
**Confidence:** 4

**Summary:**

Currently, the majority of Zero-shot TTS models predominantly focus on the aspect of audio generation quality. FlashSpeech, building upon the existing latent consistency diffusion model, achieves speech synthesis with RTF 0.02 by employing only one to two sampling steps. Furthermore, it ensures further enhancement in audio synthesis quality through two additional techniques: adversarial loss in the generator and the incorporation of a rhythm predictor introducing noise fluctuations.

**Strengths:**

1. The RTF (Real-Time Factor) of zero-shot TTS on a single V100 GPU is exceptionally low, measuring only 0.02. This outcome has surprised me.
2. Leveraging latent consistency diffusion models from other domains in zero-shot TTS, which demands high speed, is indeed an appropriate approach. Additionally, the incorporation of two simple engineering tricks, namely adversarial loss and a rhythm generator with perturbations, has led to corresponding improvements.
3. The writing is clear and comprehensible.

**Limitations:**

1. While the author has effectively packaged the introduction, FlashSpeech does not introduce any groundbreaking structural innovations. The latent consistency diffusion model is merely a direct application, and the utilization of adversarial loss and perturbed rhythm generation are engineering tricks commonly observed in the field of Speech. FlashSpeech offers some insightful implications for the speech community but lacks significant novelty.
2. It would be beneficial for the author to compare FlashSpeech with similar lightweight and fast zero-shot TTS systems, such as P-flow [1] and MobileSpeech[2].
3. I have some concerns regarding the experimental results, such as UTMOS. I have personally tested the corresponding UTMOS on encodec_gt, and the achieved performance did not reach 3.6. I am curious to know why FlashSpeech, based on encodec, surpasses encodec_gt. Could the author provide the UTMOS testing script during the rebuttal period?
4. In regard to the related ablation experiments on the Prosody Generator and Discriminator, I suggest conducting tests on a broader range of metrics, particularly for the Prosody Generator. It is challenging to demonstrate the effectiveness of the Prosody Generator solely based on Pitch JSD and Duration JSD when there is no difference in WER.
5. I find it difficult to comprehend why FlashSpeech achieves higher subjective SMOS scores than NaturalSpeech2, despite having lower Sim-O and Sim-R scores in Table 1. Additionally, there has existed numerous reproductions or open-source models in the zero-shot TTS community, as well as several new works reporting new state-of-the-art results. FlashSpeech should attempt comparisons with more baselines, as the results in Table 1 may be influenced by the gap caused by reproduction.
6. The results in Table 1 may be affected by biases arising from inconsistent training datasets. However, I understand the challenges of reproducing baselines on MLS due to the time-consuming nature of the process. I am curious as to why FlashSpeech did not consider training on librilight, as many zero-shot TTS models are trained on this dataset. Additionally, testing on LibriSpeech may have an in-domain impact on MLS, and I am interested to learn about the out-of-domain performance of FlashSpeech compared to other baseline models.

[1] P-Flow: A Fast and Data-Efficient Zero-Shot TTS through Speech Prompting

[2] MobileSpeech: A Fast and High-Fidelity Framework for Mobile Zero-Shot Text-to-Speech

**I look forward to the author addressing my concerns during the rebuttal, and I will adjust my score (high or low) based on the feedback received in the rebuttal.**

**Suitability:**

3

---

### Official Review · Reviewer_WoEh · 2024-05-12

**Rating:** 5
**Confidence:** 3

**Summary:**

This paper presents FlashSpeech, a novel speech generation system, based on the latent consistency model (LCM). Their proposed adversarial consistency training enables them to train a model from scratch without distillation. FlashSpeech achieves speeds about 20 times faster than existing TTS models, without compromising voice quality, similarity, and robustness.

**Strengths:**

- Well-designed architecture equipped with the latest technologies of deep generative modeling [53,54,32]. Although this model is made by combining existing methods, I believe publishing this paper and model will be beneficial for the community.
- Good experimental results. Competitive performance in quality with SOTA TTS models despite much faster synthesis. In addition, the proposed model is better in a use preference test. Ablation studies are also enough to convince readers.
- Well-written manuscript. I can easily understand their motivation, some details of their training framework, and experimental settings.

**Limitations:**

- Novelty is a little limited. However, as I commented in the previous field, publishing this paper and model will be beneficial for the community. So, I don't think this is a big drawback.
- I wanted to hear synthesized speech.
- Even though the manuscript is well written, it might be hard for some readers to reproduce models. By making the code and pretrained checkpoint publicly available, the authors can contribute more to the community.

**Suitability:**

2

---

### Official Review · Reviewer_X4w8 · 2024-05-24

**Rating:** 6
**Confidence:** 4

**Summary:**

This paper introduces FlashSpeech, a new text-to-speech model that delivers both low latency and high quality, as well as speaker similarity. FlashSpeech, based on the text-to-speech paradigm with duration modeling, utilizes a latent consistency model. This model helps in speech generation based on textual and prosodic features, enabling efficient few-step sampling. With the addition of an adversarial training module, the quality of the generation is improved.

FlashSpeech incorporates a prosody generator augmented with a latent consistency model. This addition of diverse residual signals onto the regressed one increases the diversity of the predicted prosody.

Comprehensive experiments were conducted by the authors to demonstrate the effectiveness of FlashSpeech.

**Strengths:**

- This paper introduces a novel text-to-speech model designed to improve sampling efficiency, while maintaining quality and speaker similarity. It effectively merges a latent consistency model with an adversarial training module to ensure both efficiency and quality. The latent consistency model greatly reduces the sampling steps, thus greatly increase the generation efficiency. The adversarial utilizing speech language model helps improve the generation quality.
- Additionally, the paper presents an innovative method for predicting prosody. This approach uses a regression model for the base and a latent consistency model for residual information, effectively capturing the diversity of prosody. A two-stage training with the first stage training the LCM and the prosody regression part and the second stage training the prosody refinement module is adopted.
- The proposed FlashSpeech proves its ability to perform voice cloning, voice conversion, speech editing, and diverse speech sampling. While the demonstrations for speech editing and diverse speech sampling are somewhat less impressive, the model's versatility is unquestionable.
- Extensive experiments are performed to assess the proposed methods and designs, including a variety of baseline methods for comparison. The effectiveness of the adversarial training module and prosody generator is clearly demonstrated through ablation studies.
- The paper is well-structured and clearly written. It introduces the background with appropriate detail and provides comprehensive explanations of key components, making it an engaging read.

**Limitations:**

- For demo of diverse speech sampling, the samples do not significantly differ in terms of pitch and duration. Specifically, one sample (the second text and sample 2) seems to have an incorrect syllable insertion (sounding like "ah e") before the word "fresh". Providing more details about the settings used to create these samples would give your readers a better understanding of the diversity you are aiming to achieve.
- While FlashSpeech's ability to facilitate speech editing is impressive, the current method appears to synthesize the new text and insert it into the editing segment. This hard concatenation results in unnatural transitions, which are noticeable in the samples from the Speech Editing section on your demo page. Perhaps exploring smoother transition techniques could enhance the naturalness of the edited speech.
- Despite using JSD for both pitch and duration to indicate their similarity to the ground-truth distribution and their diversity, it's hard to determine the accuracy of these two predicted features based solely on the JSD. Therefore, including objective metrics on pitch and duration for reference would be beneficial.

**Suitability:**

2

---

### Official Review · Reviewer_4FCQ · 2024-05-24

**Rating:** 4
**Confidence:** 3

**Summary:**

FlashSpeech introduces latent consistency model and adversarial consistency training that significantly reduces inference time, achieving 20 fold speed increase while maintaining high speech quality and speaker similarity. The paper presents a significant advancement in zero-shot speech synthesis.  It supports diverse tasks like voice conversion, speech editing, and diverse speech sampling.

**Strengths:**

(1) The proposed architecture achieves up to 20 times faster inference than previous models, leveraging a non-autoregressive framework, Latent Consistency Model, and adversarial consistency training. This efficiency is complemented by high-quality speech synthesis, espicailly in zero-shot speech synthesis senarios.

(2) High-quality audio output and strong speaker similarity in zero-shot scenarios.

(3)  Broad applicability of the proposed architecuture across various tasks such as TTS, voice conversion, speech editing etc. highlights the model's robustness and potential for real-world deployment in multiple speech synthesis scenarios.

(4) The paper provides extensive experimental results that clearly demonstrate advantages over existing methods.

**Limitations:**

(1) Limited novelty in the architecture.FlashSpeech replicates the core architecture of NS2 but integrates a consistency model to enhance performance.

(2) The paper does not include an evaluation against StyleTTS2 [1], a method known for achieving comparable quality with significantly less data. Including StyleTTS2 in the evaluation would also help assess the generalizability of FlashSpeech.

(3) To improve the reader's understanding, Figure 2 in the paper should be updated to clearly distinguish between trainable and non-trainable parameters.

(4) The Statistical Significance analysis for the subjective evaluation is missing. To ensure the reliability of the subjective evaluation, it is crucial to perform a statistical significance analysis to demonstrate that whether the observed differences results between FlashSpeech and other baselines are statistically significant.


[1] Li, Yinghao Aaron, et al. "Styletts 2: Towards human-level text-to-speech through style diffusion and adversarial training with large speech language models." Advances in Neural Information Processing Systems 36 (2024).

Minor Comments:

-How the model will perform with noisy or highly variable real-world data? Are there any robustness measures to ensure consistent performance across different acoustic environments and speaker variations?

-Some discussion on how prosody generator balance the contributions of deterministic and stochastic components in the prosody refinement process is required.

Missing Citation:  The previous uses of the Latent Consistency Model (LCM) in Text-to-Speech (TTS) systems might be relevant.
Li, Xiang, et al. "CM-TTS: Enhancing Real Time Text-to-Speech Synthesis Efficiency through Weighted Samplers and Consistency Models." In Findings of NAACL, 2024.

**Suitability:**

3

---

### Meta-Review · Area_Chair_UtzB · 2024-07-02

**Recommendation:** Accept (Poster)
**Confidence:** 4

**Metareview:**

This paper introduces FlashSpeech, a large-scale zero-shot speech synthesis system with efficient inference and high quality outputs. It is  built on the latent consistency model and uses a new prosody generator module to generate more natural sounding outputs. It has been shown to be effective for a broad range of tasks such as voice conversion, speech editing, and diverse speech sampling.

Strengths:
- The model is quite diverse and does well on a multitude of tasks which is quite impressive.
- The linked audio samples are of great quality and a testament of this work.
- Very well-written paper with a good amount of focus on experimental results.
- The choice of model components has been explained well and seems to resonate with most reviewers.
- Open sourcing the codebase would be a great way of advancing research in this area of work.

Weaknesses:
- It has been noted in multiple reviews that there isn't much innovation in modeling itself.
- Comparison with similar, recent TTS systems is missing and would be a valuable addition to the discussions in the paper.